# The Formation of a Rubble Pile Asteroid: Insights from the Asteroid Ryugu

**Tsutomu Ota \***, **Christian Potiszil**, **Katsura Kobayashi**, **Ryoji Tanaka**, **Hiroshi Kitagawa**, **Tak Kunihiro**, **Chie Sakaguchi**, **Masahiro Yamanaka** **and Eizo Nakamura**

Pheasant Memorial Laboratory, Institute for Planetary Materials, Okayama University, Yamada 827, Misasa 682-0193, Tottori, Japan
* **Correspondence:** ohta-t@misasa.okayama-u.ac.jp

**Abstract:** The Hayabusa2 mission returned primitive samples from the C-type asteroid Ryugu to Earth. The C-type asteroids hold clues to the origin of Earth's water and the building blocks of life. The rubble pile structure of C-type asteroids is a crucial physical feature relating to their origin and evolution. A rubble pile asteroid is hypothesized to be bound primarily by self-gravity with a significant void space among irregularly shaped materials after catastrophic impacts between larger asteroids. However, the geological observations from Hayabusa2 and the analyses of the returned sample from Ryugu revealed that the high microporosity was common to various >10 m- to mm-sized materials of Ryugu, which suggests that the asteroid Ryugu is not just a loosely bound agglomeration of massive rocky debris from shattered asteroids. For a better understanding of the origin and evolution of the rubble pile asteroid, the current most accepted hypothesis should be verified by observations and laboratory analyses and improved upon based on this information. Here, the previous models are examined using Hayabusa2's geological observations of the asteroid and the analytical data from the samples returned from Ryugu's surface and subsurface material. Incorporating the new findings, a hypothesis for the evolution of the rubble pile asteroid Ryugu from a cometary nucleus through sublimation and subsequent dynamic resurfacing is proposed. The proposed hypothesis is applicable to other rubble-pile asteroids and would provide perspectives for near-Earth objects in general.

**Keywords:** rubble pile; asteroid; comet; sublimation; Ryugu

## 1. Introduction

Asteroids and comets are the materials that were left over after the formation of the Sun and the planets and that would have been formed from gas and dust in the protosolar nebula (PSN). Accordingly, such materials could have preserved clues about the processes during the early period of the solar system. The PSN would have been spinning faster towards its center, which would have concentrated the materials within this region. Some of the materials would have fallen into the protosun and increased its temperature. Through the increased radiation emitted by the hotter protosun within the inner solar system, thermally evolved materials would have eventually formed S-type asteroids [1]. Meanwhile, C-type asteroids such as Ryugu, targeted by the Hayabusa2 mission [2], have spectra similar to those of carbonaceous chondrite meteorites [1] and are expected to preserve far more materials from the primitive outer solar system. Thus C-type asteroids may hold clues concerning the origin of Earth's water and the building blocks of life.

Remote sensing observations from the Hayabusa2 spacecraft revealed that Ryugu was a spinning top-shaped rubble-pile asteroid [3,4]. Near-infrared spectroscopy found ubiquitous hydrated minerals and a low albedo for the surface of Ryugu [5]. Since the albedo of a solar system body is affected by the abundance of organic matter and minerals

present on its surface, it is possible to determine the abundance of organic matter required to explain the low albedo of Ryugu's surface through a series of mass balances involving irradiated organic matter and mineral components. Accordingly, mass-balance calculations based on remote sensing data and observations concerning the albedo of material displaced after the touchdown of the Hayabusa2 spacecraft showed that the surface of Ryugu could contain a higher abundance of organic matter than that in carbonaceous chondrites [6]. The potential for a high abundance of organic matter and the rubble-pile nature of Ryugu motivated the numerical modeling of the sublimation of water ice from a cometary nucleus, which could leave behind lithic debris and organic matter as evaporative residues [7].

Recent space missions to asteroids Itokawa, Ryugu, Bennu, and Dimorphos, and progress in numerical models indicate that rubble-pile asteroids are more common than previously thought in the main asteroid belt (e.g., [8]). A rubble pile asteroid is hypothesized to consist of reassembled macroscopic materials held together by their self-gravity after catastrophic impacts between larger asteroids (e.g., [9]). However, observations of the asteroid Ryugu by Hayabusa2 demonstrated not only that this asteroid was a low-density rubble pile with large boulders on its surface (e.g., [3]), but also that the large boulders (>10 m) and cm-sized pebbles on Ryugu had high microporosity [10,11], which is consistent with the overall low bulk density ($1190 \pm 20$ kg·m$^{-3}$) and a high bulk porosity (30~50%) [3,4]. Furthermore, boulders with a high porosity of >70%, which is higher than Ryugu's average and as high as in cometary bodies, have also been discovered [12]. The high microporosity common to various-sized materials of Ryugu suggests that asteroid Ryugu is not just a loosely bound agglomeration of massive rocky debris from shattered asteroids.

The bulk porosity (or macroporosity, resulting from void spaces among the constituent boulders) of a rubble pile asteroid is the result of processes that occurred during the evolution of a progenitor planetesimal, including its formation, destruction, and the subsequent arrangement of the rubble pile asteroid's constituent materials. On the other hand, the porosity of a boulder is a local microporosity (relevant to void spaces within the boulder) resulting mainly from processes that took place during the thermal evolution of a progenitor planetesimal from which the boulder originated prior to the formation of a rubble pile asteroid. Nevertheless, the internal structure of a rubble pile asteroid, namely, how the voids are distributed and how the materials are interconnected to achieve high porosity, is not very constrained. In addition, the collisional lifetime of 0.2~10 km sized rubble-pile asteroids is considered to be short compared to the age of the solar system (e.g., [9,13]), but it has recently been discovered that the dust particles, retrieved from the <1 km sized, near-Earth asteroid Itokawa, indicate that the age of formation of the rubble pile structure is greater than 4.2 Ga [8]. Such a long survival time against the ambient bombardment in the inner solar system, namely the resistance to collisions and fragmentations, is likely caused by the shock-absorbent nature of the highly porous rubble-pile asteroid (e.g., [8,14]). For a better understanding of the origin and evolution of a rubble pile asteroid, the numerical models with hypothetical parameters should be verified using the information from the on-site observation and laboratory analysis of the retrieved sample if available.

While the laboratory analysis of uncontaminated primitive samples was highly anticipated, the Hayabusa2 spacecraft collected samples of Ryugu's surface and subsurface materials from the first touchdown site (TD1) and the second touchdown site (TD2), respectively [15–17], and brought the samples to the Earth on 6 December 2020. Around 5.4 g of the sample was returned and studied during the overall preliminary analysis at the phase-1 curation (P1C) facility [18]. Subsequently, 16 particles of the Ryugu samples (1.2~3.7 mm in diameter and ~55 mg in total) were allocated to the Pheasant Memorial Laboratory (PML) at the Institute for Planetary Materials, Okayama University at Misasa, designated as a phase-2 curation (P2C) facility [19]. Here, we summarize the comprehensive laboratory analyses of the returned samples with the Hayabusa2 geological observations on the asteroid's surface, with a focus on those performed at the P2C facility of PML, and verify whether the previously proposed models fully explain the results of the laboratory analyses and the observations. Finally, we propose one plausible scenario for the formation of an

icy planetesimal and its evolution to the rubble-pile asteroid Ryugu, with implications for near-Earth rubble-pile objects.

## 2. The Nature of Ryugu Particles

The main unique feature of the Ryugu particles was their distinctively low bulk density of $1528 \pm 242$ kg·m$^{-3}$, with particle-by-particle variations from 1200 to 2080 kg·m$^{-3}$ [19], which cover the average bulk density of Ryugu, namely $1190 \pm 20$ kg·m$^{-3}$ [3], and the bulk densities of another batch of Ryugu particles, $1790 \pm 80$ kg·m$^{-3}$ estimated by considering their 3D structure [20], and overlap the bulk densities of Ryugu particles from the preliminary analysis of the P1C, $1282 \pm 231$ kg·m$^{-3}$ [18]. The low reflectance and low bulk density, and the resultant high porosity of the Ryugu particles, were comparable to the ungrouped carbonaceous chondrite, Tagish Lake (Table 1 in [18]), which is considered to have originated from a D-type asteroid characterized by a very low albedo [21,22]. However, the Ryugu particles were found to be similar to CI chondrites in terms of their petrology, mineralogy, and geochemistry, being heavily affected by aqueous alteration (e.g., [19,20,23]).

The bulk elemental and isotopic abundances, and organic carbon contents (1.8∼4.0 wt.%) were in agreement with those of CI chondrites [19,23–26]. The O-Cr isotopic compositions [19,23,27] indicated that Ryugu sampled the least thermally processed PSN material. The Ti-Fe isotopic compositions suggested their formation in the outer solar system around the birthplaces of Uranus and Neptune, namely ∼13 to 25 astronomical units (AU) from the Sun [23,28]. The Ne isotopic compositions indicate that Ryugu particles contain solar-wind Ne that was recently implanted, particularly in TD1 particles on the surface of Ryugu [19,29]. However, the TD2 particles from the subsurface were found to have inherited ancient solar-wind Ne [19], making it difficult to obtain reliable cosmic ray exposure ages.

The Ryugu particles were composed of 50% matrix and 9% coarse-grained components, with an average porosity of 41% [19]. In the matrix, the phyllosilicate minerals, which were the inter-layered Al-rich smectite-group and the Fe-rich serpentine-group minerals formed through aqueous alteration, were associated with sub-μm to μm sized components including appreciable amounts of organic matter (OM), Fe sulfide, magnetite, carbonate, and phosphate minerals, as well as numerous voids (Figure 4 in [19]). The presence of $CO_2$-bearing fluid inclusion in a Fe sulfide crystal indicated that its crystallization occurred beyond the $H_2O$ and $CO_2$ snow lines of the early solar system >3 to 4 AU from the Sun [20], which is consistent with the bulk Ti-Fe isotopic compositions and suggests that the material in the Ryugu particles accreted in the outer solar system [23,28].

The ubiquitous OM in the matrix indicated by micro-Raman spectroscopy and micro-Fourier transform infrared spectroscopy suggested the presence of sub-μm- to nm-sized insoluble OM (IOM) throughout the matrix [19,30]. The bulk C-N isotopic compositions were controlled by the abundance of IOM, characterized by the H- and N-isotope anomalies compatible with origins of both PSN and interstellar medium (ISM) [19,30]. Such isotope anomalies have also been found in CI and CM chondrites, as well as in the Tagish Lake meteorite comparable to D-type asteroids, interplanetary dust particles, and comets (e.g., [31–33]). The N contents (expressed as N/C atomic ratios) of the IOM in the Ryugu particles, 0.010∼0.035 [30], which is consistent with the bulk N content of 0.10∼0.22 wt.% [19,25], were found to be lower than those of OM in the Tagish Lake meteorite, 0.04 [32], ultracarbonaceous micrometeorites, 0.05∼0.15 [34], and the comet 81P/Wild 2, 0.08∼0.16 [33]. However, the N contents of dust particles from the comet 67P/Churyumov-Gerasimenko are heterogeneous (0.02∼0.06 in N/C) and overlap with the values of IOM in CM, CI, and CR chondrites, 0.03∼0.04 [35] and the Ryugu particles, which could be attributed to the presence of semivolatile ammonium salts as a substantial N reservoir in this comet (e.g., [36]). The MicrOmega hyperspectral microscope detected NH-rich compounds such as $NH_4$-phyllosilicates, $NH_4$-hydrated salts, and N-rich OM, from the Ryugu particles [37], and formation pathways involving ammonia were proposed to synthesize the OM, including a number of amino acids, detected within the Ryugu

particles [19,25,38,39]. As such, the low N/C ratios of the Ryugu IOM would not contradict the accretion of N-bearing phases in the cold outer solar system into the Ryugu progenitor planetesimal.

Meanwhile, the spatial data for soluble OM (SOM) components and phyllosilicate minerals from both Ryugu and carbonaceous chondrites suggest that during aqueous alteration on a progenitor body, the SOM that migrated with aqueous fluid was adsorbed onto the surface or into the interlayers of the resultant phyllosilicate minerals [19,25,30,40]. As such, the SOM, including complex *α*- and *β*-amino acids that were synthesized through the interaction with the Al-rich smectites in the Ryugu progenitor planetesimal [19,38], would have been spread ubiquitously in the phyllosilicate-dominated matrix, and together with the widespread IOM, might cause the low albedo of the surface of the current Ryugu. Additionally, the detection of a wide variety of organic molecules including amino acids, such as those found within the proteins of living organism on Earth, in the terrestrially uncontaminated Ryugu particles [19,25,30,38,39,41–43] proved an indigenous origin for such amino acids in meteorites.

The coarse-grained components were 10's of μm to 100 μm in size, and included nodular aggregates of phyllosilicate, carbonate, phosphate, Fe-sulfide, magnetite, and OM (e.g., Figure 3a in [19]), and grains of olivine, low-Ca pyroxene, and phosphate. Some grains of olivine and pyroxene were found to be partially replaced by phyllosilicates, which suggests that those grains have been indigenous to Ryugu and affected by aqueous alteration [44]. In situ analyses of O isotopes in unaltered olivine and low-Ca pyroxene suggested that these minerals were derived from exogenous materials (Figure 3e,f in [19]), such as the chondrules of carbonaceous chondrites and the amoeboid olivine aggregates (AOAs) of refractory inclusions (e.g., [45,46]). Like these unaltered phases that have been incorporated after aqueous alteration [19], fragments of chondrules and Ca–Al-rich inclusions (CAIs) that experienced high temperatures in the early solar nebula were found to be rare and small ($\leq$10$\sim$30 μm in size) among the coarse-grained components in the Ryugu particles [19,20,27,44,47,48].

## 3. Formation of a Progenitor Planetesimal

The asteroid Ryugu is thought to have been an icy planetesimal that formed through the accretion of dust particles, consisting of GEMS (glass with embedded metal and sulfides) [49]-like matter and ice mantles that contain OM molecules. The accreted components recorded various origins, including the ISM and PSN. The findings from the laboratory analyses, including the abundant water and very little, thermally evolved, inner solar system material, suggest that the material in the Ryugu particles was accreted in the outer solar system at >3 to 4 AU [20], or possibly $\sim$13 to 25 AU from the Sun, where is close to the accretion region of Oort cloud comets [28].

The Ryugu particles were predominantly composed of hydrous phyllosilicate minerals, suggesting a pervasive aqueous alteration (Figure 1). The crystallization temperature for carbonate and magnetite pairs during aqueous alteration was estimated to be $\approx$30 °C, based on O isotope thermometry [19,23], and consistent with the degree of degradation of OM [27] and the mineralogical features of Fe-sulfide phases [20]. The physical modeling to simulate the conditions reproducing the mineral assemblage and the modal abundances of current Ryugu suggested that a water-to-rock ratio of 0.5–1.0 was required for a progenitor body of radius 28–40 km [19]. This water-to-rock ratio was found to be consistent with the water-to-rock ratio of $\geq$0.7, estimated on the assumption that the voids, which defined the porosity of Ryugu ($\approx$41%) [19] were fully occupied by water ice.

The $^{53}$Mn-$^{53}$Cr dating of carbonates suggested that the aqueous alteration would have peaked before $\approx$ 3 Myr after the formation of CAI, with certain uncertainties depending on the samples examined and the analytical details ($2.6^{+1.0}_{-0.8}$ [19], $5.2^{+1.6}_{-2.1}$ [23], and $\leq$1.8 Myr [50]). This means that the materials from Ryugu experienced liquid water at a very early stage of the solar system's history, and the heat that melted the ice could have been supplied from radioactive isotopes such as $^{26}$Al that only survive for a few Myr. To form liquid water

from the heating of a rocky-icy body by radioactive decay, the body must be at least several 10's of km in size [51], which is consistent with the size of the progenitor body predicted by the aforementioned physical modeling [19]. Accordingly, the icy planetesimal would have been a much bigger body than the current asteroid Ryugu.

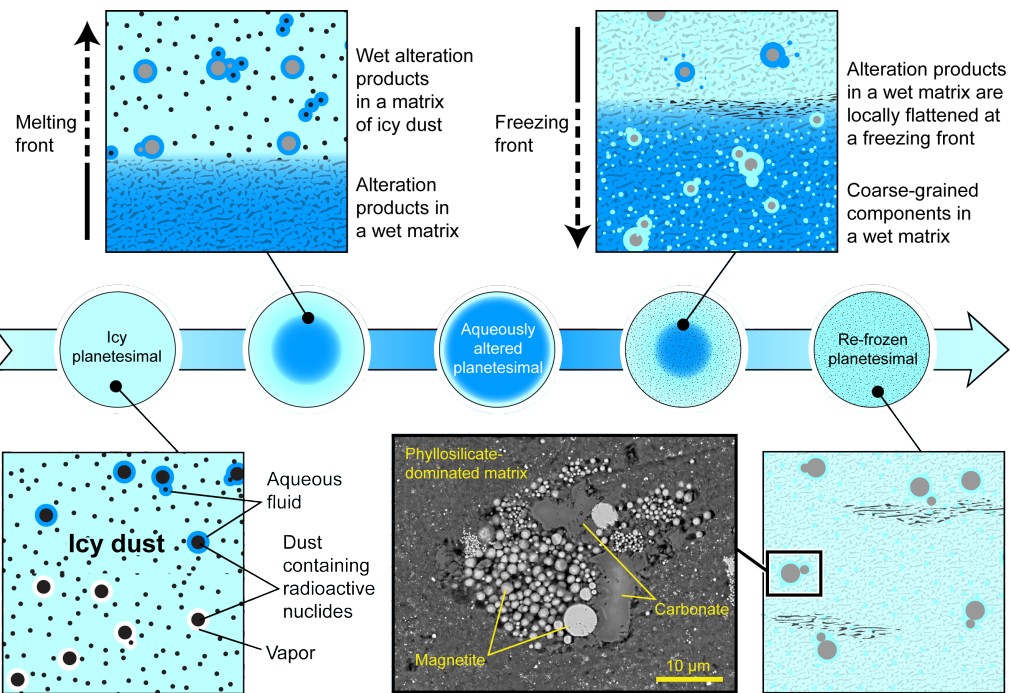

**Figure 1.** Aqueous fluid-related processes in an icy planetesimal deduced from the Ryugu particles. (**Bottom left**) In a planetesimal composed of icy dust, the heat generated by the decay of radionuclides causes a phase transition of the ice around the dust. The ice sublimates under extremely low-pressure conditions near the surface, but under lithostatic pressure in the interior, it transforms into liquid water. (**Top left**) Silicate components with organic matter in the dust are converted into hydrous silicate minerals by reactions with aqueous fluid. The aqueous alteration proceeds from the inside of the planetesimal toward the surface, and the volume of the wet matrix with alteration products increases. (**Top right**) As the decay of radionuclides is terminated, the aqueously altered planetesimal begins to freeze from the surface. Due to the difference in freezing point and reaction rate related to the diversity in chemical composition of the aqueous fluid, a transition zone consisting of solid-liquid composites develops near the boundary between the wet and the icy matrices and migrates toward the interior, where coarse-grained components are grown through freezing-thawing cycles. The alteration products, loosely distributed in the wet matrix, are pressed by the re-frozen icy matrix that grows inward due to the volume change accompanying freezing, and are locally flattened at the freezing front (e.g., Figure 6a in [19]). (**Bottom right**) During the planetesimal re-freezing, the droplets of aqueous fluid that are dispersed between large quantities of minerals and organic matter components, repeatedly freeze and thaw in isolation from the earlier frozen matrix. Finally, they become coarse-grained components such as the carbonate nodule, accompanied with magnetites with different crystal habits and sizes, found in the Ryugu particle (A0048-3, inset).

After significant abundances of radioactive isotopes had decayed to induce the aqueous alteration producing a vast amount of phyllosilicates, the icy planetesimal would have cooled, but with minor perturbations in temperature, and frozen again (Figure 1). During freeze-thaw cycles by the temperature perturbations at this cooling stage, ice crystals could have nucleated and grown, pushing the matrix constituents out of the way and compressing them. As the ice melted, an open space filled with aqueous fluid would have been formed where nucleation and growth of magnetite, carbonate, and inter-crystalline phyllosilicate could have occurred (Figures 5d and 6d in [19]). In this way, the complex textures of the

coarse-grained components, such as magnetite-carbonate nodules (Figure 1), could have been formed at the ∼10 μm-scale [19].

The icy planetesimal could have been disrupted and broken up by an impact, after it had re-frozen in the outer solar system. In such a scenario, a km-sized fragment of the icy planetesimal, which preserves many of the original textures and physicochemical properties of the original planetesimal as stated above, could be transferred into the asteroid belt due to interactions with the terrestrial planets (e.g., [52]).

## 4. Evolution to a Rubble Pile Asteroid

Rubble pile asteroids, which are characterized by their low density, have been suggested to form when an asteroid is shattered into pieces by an impact, and the pieces subsequently come back together, primarily due to self-gravitation (e.g., [9,13,53]). The intense impact required to shatter an asteroid into pieces may generate high pressure and temperature at the impact site and the interior of the asteroid. Although a large number of impact craters were observed on the surface of current Ryugu [4,16], no evidence of such extreme conditions has been recognized, and low-temperature products such as phyllosilicates and OM are abundantly preserved in the Ryugu samples investigated so far [19,20,23,27,30]. The recently discovered microfault-like textures and the high-pressure polymorph of Fe-Cr sulfide in the Ryugu particles indicated that asteroid Ryugu has experienced mild shocks with the average peak pressure of <2 GPa [54], but this is unlikely to support an impact capable of shattering the Ryugu progenitor body. Alternatively, the high microporosity of the progenitor body, similar to that of a yarn ball, which consists mainly of phyllosilicates interconnected with nm- to μm-sized voids, might have precluded any record of intense impacts on Ryugu.

Accordingly, a head-on collision of a 12-km-sized impactor into a 100-km-sized Ryugu progenitor body with the physical properties measured from the Ryugu samples was simulated [20]. The resulting temperature did not rise above 100 °C at regions 10∼25-km away from the impact site. Thus, Ryugu might have formed from fragments excavated from regions far from the impact site [20]. However, the gravitational reaccumulation of the fragments after the impact does not explain the important geological characteristics of current Ryugu, namely, why the physicochemical properties of the TD1 surface material differ from those of the TD2 subsurface material. Some of the TD1 samples collected from the surface of Ryugu show elemental fractionation beyond the mm scale and scattered elemental abundances of B, which is highly fluid-mobile. However, all TD2 samples from the subsurface record elemental abundances similar to CI chondrites and show no evidence of fluid-related elemental fractionation over the mm scale [19].

Instead, a cometary nucleus, composed of rock, water ice, and frozen gases, is also characterized by low density (e.g., [9]). The km-sized, icy planetesimal fragment, as a cometary nucleus-like body, would have moved from the outer to the inner solar system by some dynamical pathway, involving the interactions of the planets in the early solar system (e.g., Nice model) [55–57]. Once in the inner solar system, the icy planetesimal fragment as the Ryugu progenitor body would have undergone significant sublimation (Figure 2).

Physical modeling in a previous study [7] indicated that, with the complete sublimation of water ice, a cometary nucleus-like body with an initial radius of 1.2 km would shrink to a rubble-pile asteroid with a radius approximately equal to that of current Ryugu (≈420 m). The subsurface voids left after the sublimation would have caused the collapse of upper roof blocks, as well as the resultant rubble pile structure (Figure 2). The collapsed blocks would have conserved the angular momentum on the whole. Coupled with the reduction in the size of the cometary nucleus-like body through the sublimation, the angular velocity could have increased, causing the spinning top shape. The sublimation could have produced water vapor jets, as seen on the comet 67P/Churyumov-Gerasimenko (e.g., [58,59]), which suggests that a cometary nucleus may form a rubble pile through sublimation, including jets. Around the pits created by the jets, subsurface materials with

low velocities below the escape velocity within the jet would have redeposited onto the surface and become frozen in place, creating conformable strata (Figure 2).

The pits observed on the comet 67P are 50∼310 m in diameter, with depth-to-diameter ratios of 0.11∼0.92, and the cumulative boulder-size distribution is theoretically consistent with the boulders in the pits being debris that fell from the pit walls [59]. The sizes of craters (10∼300 m) and rubble materials (a few cm to 300 m) observed on the surface of Ryugu [4,16,17] are comparable in size to the pits on the comet 67P. These observations imply that some of the craters and the rubble material on Ryugu could have formed as a result of sublimation through jets.

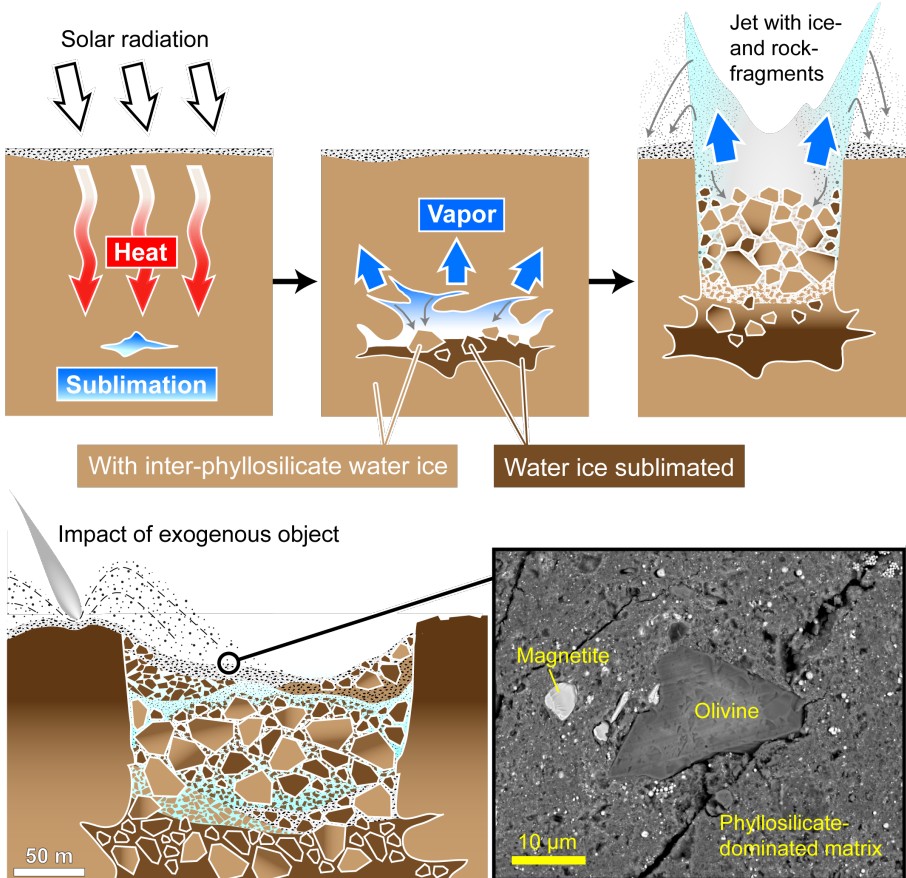

**Figure 2.** The dynamical evolution of the cometary nucleus-like body that would become the rubble-pile asteroid Ryugu. (**Top**) As a cometary nucleus-like body moves towards the inner solar system, solar radiation causes subsurface ices to sublimate, forming cavities. Subsequent sublimation and water vapor pressure propagate the fissures and enlarge the cavities, eventually causing the ceiling to collapse. The collapse forms a pit resembling a sinkhole and exposes fresh materials in the walls of the pit, which sublimate to produce water vapor jets, accompanied by subsurface materials. Parts of the debris are redeposited on the surface, and are sintered together with surface materials by refreezing the water vapor in place. (**Bottom**) During the repeated sublimation and resurfacing, the debris deposits above the subsurface materials collapsed into the pit, along with the surface materials containing fragments from exogenous objects around the pits, formed a conformable stratum, and intermingled with the materials with and without inter-phyllosilicate water ice. As such, the surface of a cometary nucleus-like body was modified locally, resulting in the formation of a rubble pile structure, and the implantation of the exogenous materials such as the unaltered olivine grain (A0073-5, inset) and the massive domain accompanying the foliated matrix (Figure 1e in [19]).

Most importantly, the jets would likely be able to explain the aforementioned differences between the sampling sites where the Ryugu samples were obtained. One explanation is that the TD1 site records the material entrained in a jet, brought to the surface of a cometary nucleus-like body from many distinct regions of the subsurface, and thus represents a wide variety of compositions. Meanwhile, the TD2 samples may represent material sourced from one part of the body and as such have a more uniform composition. Subsequently, through extensive sublimation of the ice, a low-density and highly porous rocky asteroid, with a rubble-pile structure, was formed (Figure 2).

For the timing of the sublimation, the systematics of highly siderophile elements (HSEs) and the non-isochronous Re-Os isotopes of Ryugu particles [19] may provide evidence that it was a geologically recent (≪1 Ga) event (see Figure A1 for details). Upon sublimation, the jets of water-rich vapors, which contain highly reactive gas components (e.g., $H_2S$, HCl, and HF) [20,60], could have reacted with the sub-µm-sized HSE-bearing compounds such as Fe sulfide in the matrix, resulting in a selective sublimation of more volatile Os-bearing molecules (Figure 3). If this is the case, the non-isochronous behavior of the Re-Os isotopes found in carbonaceous chondrites [61] may have resulted from sublimation on their icy progenitor bodies.

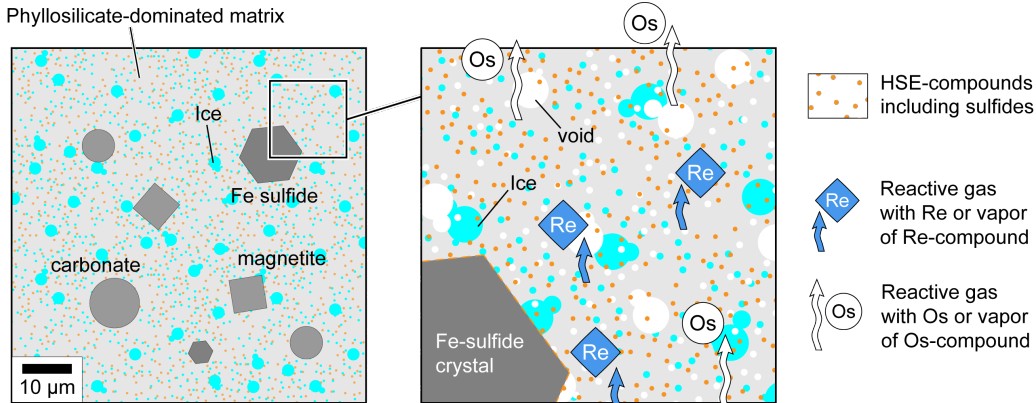

**Figure 3.** The sublimation-induced Re-Os fractionation model deduced from the Ryugu particles. (**Left**) The HSE fractionation between coarse-grained Fe sulfide crystals and fluid during the aqueous alteration would have resulted in the enrichments of Re, Ir, and Pd relative to Os, Pt, and Ru in the fluid. The HSE-bearing fluid would have precipitated as the nm-sized HSE compounds in the matrix or frozen as impurities in the µm-sized water ice. (**Right**) The sublimation of water ice with jets (Figure 2) would have generated reactive gases, which would have subsequently dissolved the nm-sized HSE compounds. Alternatively, the nm-sized HSE compounds in the water ice could have directly sublimated together with the ice. More volatile HSE compounds such as $OsO_4$ and $RuO_4$ could have been preferentially sublimated, but the other HSE compounds were left in the voids within the matrix left after the ice sublimated.

## 5. Implications for Near-Earth Objects

The interrelationships between comets and asteroids have long been debated (e.g., [52,62–65]). A cometary origin for the CI1 chondrite, Orgueil, was proposed based on cosmochemical and astronomical data, including a critical examination of the Orgueil orbit using ∼160-year-old visual observations [66]. As a unique example, the interaction between an asteroid fragment and a cometary nucleus decoded from the Chelyabinsk ordinary chondrite was argued [67]. While telescopic observation and theoretical simulation indirectly suggest the evolution of comets into asteroids, the information directly extracted in terms of material science through the comprehensive laboratory analyses of the Ryugu samples and the Hayabusa2 geological observations of the current Ryugu surface was compatible with an origin via a cometary nucleus-like body that evolved through sublimation and subsequent resurfacing to yield a dormant comet and thus a C-type rubble pile asteroid.

The numerical simulation for the sublimation of water ice from a cometary nucleus-like body, accompanied by the spin-up to shrink the nucleus [7], indicated that the cometary origin scenario was able to account for the distinctive features of the current Ryugu. In this simulation, an initial cometary nucleus is composed of μm-sized water ice particles and cm-sized rocky debris. The water ice sublimates from the outer layer of the nucleus, and the remaining rocky debris piles up on the surface to form a dust mantle, which grows steadily inward. If these hypothetical parameters and assumptions are contradictory to the practical observation and the analytical results of the sample returned, it will be possible to influence the contradiction in the simulation and to improve the scenario by incorporating the new findings.

In fact, the physicochemical properties of Ryugu revealed by the laboratory analyses suggested that the rubble material on Ryugu would not have been massive rocky debris, but rather porous objects, composed of the water ice mixed with a phyllosilicate-dominated matrix at the nm to μm scale. The Hayabusa2 observation has indicated that the large (>10 m) boulders on the surface of Ryugu exhibited low thermal conductivities and accordingly have high porosities (28∼55%) [10], which were found to be comparable to the microporosity estimated from the mm-sized Ryugu particles (≈41%) [19].

The sublimation would have been closely related to the jets from the ∼100 m-sized pits as observed on the comet 67P, and caused the collapse of materials near the surface, resulting in the formation of rubble material of various sizes in the pits, which are composed of the nm- to μm-sized water ice particles with a phyllosilicate-dominated matrix. The subsurface material entrained in the jet would have been brought to the surface of the cometary nucleus-like progenitor body, redeposited as the porous debris, and sintered together with surface materials by refreezing of the water vapor. Such a dynamical process of sublimation involving jets and the resultant resurfacing is crucial for producing a rubble-pile asteroid.

The hypothesis proposed here can explain the loss of volatiles from the Ryugu progenitor body, the dynamical evolution of Ryugu, the low albedo of the surface of current Ryugu, and the physicochemical properties of the Ryugu samples. This hypothesis is also applicable to other rubble pile asteroids, such as Bennu, targeted by the OSIRIS-REx mission [68–71]. Furthermore, the physicochemical properties of Ryugu and its progenitor deduced from the information extracted might provide perspectives for near-Earth objects in general. Using the materials with densities equivalent to the Ryugu particles (∼1530 kg·m$^{-3}$) [19], approximately 22% of macroporosity is required to achieve the bulk density of the asteroid Ryugu (1190 ± 20 kg·m$^{-3}$) [3]. Such a high macroporosity suggests that the highly porous materials, which would have been less processed in the progenitor planetesimal [12], survived more abundantly in the subsurface. Alternatively, the current Ryugu may represent a dormant cometary nucleus-like body containing water ice within its interior [19], which may be able to explain why C-type asteroids are less dense than the meteorites that are thought to be formed from them.

## 6. Future Directions

In this review, we summarized the recent laboratory analyses of the returned samples with the Hayabusa2 geological observations of asteroid Ryugu, and proposed a plausible hypothesis for a cometary nucleus-like body origin and the sublimation-driven evolution of the rubble-pile asteroid Ryugu, in the context of the Rosetta OSIRIS observation of the comet 67P. The hypothesis proposed here has taken into account the new findings contradictory to the hypothetical parameters and assumptions in previous models, but there is still room to improve it by theoretically evaluating the influence based on the contradictions.

Future numerical simulations, using the analytical data acquired from the Ryugu samples and the geological remote sensing observations of the surface of Ryugu, must help to further elucidate realistic scenarios for the dynamical evolution of rubble-pile asteroids. In particular, based on the work outlined in this review, the transformation from a comet to an asteroid should be examined in detail. Indeed, it was suggested that asteroids and comets belong to a general population of small solar system bodies that exhibit a continuous range of physical and chemical properties (e.g., [52,65]).

The cometary nucleus origin for Ryugu, which was proposed here, explains how a cometary nucleus transforms to a rubble pile asteroid, but the time evolution of the radius of the cometary nucleus and the duration of ice sublimation depend on different parameters [7]. The initial water (ice)-to-rock ratio of a cometary nucleus concerns the contraction rate of the nucleus due to ice sublimation, and its resultant spin-up causes the spinning top shape. The duration of ice sublimation depends on the macroporosity and the size of the rocky debris of the cometary nucleus. The microporosity of rubble material and episodic sublimation involving the jets would also affect the sublimation rate of ice in the cometary nucleus. As such, numerical simulations adopting the parameters deduced from the recent findings should be performed in order to elucidate realistic, spatiotemporally constrained scenarios for the evolution of rubble-pile near-Earth objects.

Recently, NASA's planetary defense mission, the Double Asteroid Redirection Test (DART) (e.g., [72]) successfully crashed a spacecraft into Dimorphos, a moonlet of the Didymos–Dimorphos binary asteroid system. The DART's onboard imager revealed the surface of Dimorphos to be covered by a large number of boulders of various sizes [73,74], and the Hubble Space Telescope captured the comet-like tails of the asteroid Dimorphos developed after the DART impact [75,76]. Further, a network of citizen scientists' telescopes found the reddening of the ejecta on the impact, which may be due to irradiation of organics [77], as seen on short-period comets (e.g., [78,79]). These findings imply that Dimorphos is a rubble pile, and may be a dormant comet, even though the Didymos-Dimorphos system is classified as an S-type asteroid (e.g., [80–82]).

The coming missions, including the OSIRIS-REx to asteroid Bennu (e.g., [68]) and the ESA's Hera to the binary asteroid Didymos–Dimorphos (e.g., [83]), would provide another set of invaluable materials and observations for understanding the origin and evolution of rubble-pile near-Earth objects. The evaluation of the hypotheses so far proposed by the new findings from such missions will enable a better understanding of the formation and evolution of our solar system.

**Author Contributions:** Conceptualization, E.N.; writing—original draft preparation, T.O.; writing—reviewing and editing, T.O., C.P., R.T., H.K., E.N., T.K., K.K., C.S. and M.Y.; visualization, T.O., K.K. and R.T. All authors have read and agreed to the published version of the manuscript.

**Funding:** This study was supported by the programs for "Promoting the Enhancement of Research Universities" and "National University Innovation Creation" to Okayama University by the MEXT and the Cabinet Office of Japan, respectively.

**Data Availability Statement:** The original images and data in all figures are available in an article [19] published open access (CC BY-NC 4.0).

**Acknowledgments:** We deeply appreciate the Hayabusa2 project team for the successful sample-return mission. We thank the PML members for their cooperation in the laboratory work on Ryugu particles and the maintenance of the laboratory. Discussions with Hitoshi Miura prior to the analysis of the samples returned were very helpful in understanding the dynamics concerning the formation of the asteroid Ryugu.

**Conflicts of Interest:** The authors declare no conflict of interest.

## Appendix A. The Systematics of Highly Siderophile Elements and Their Isotopes in the Ryugu Particles

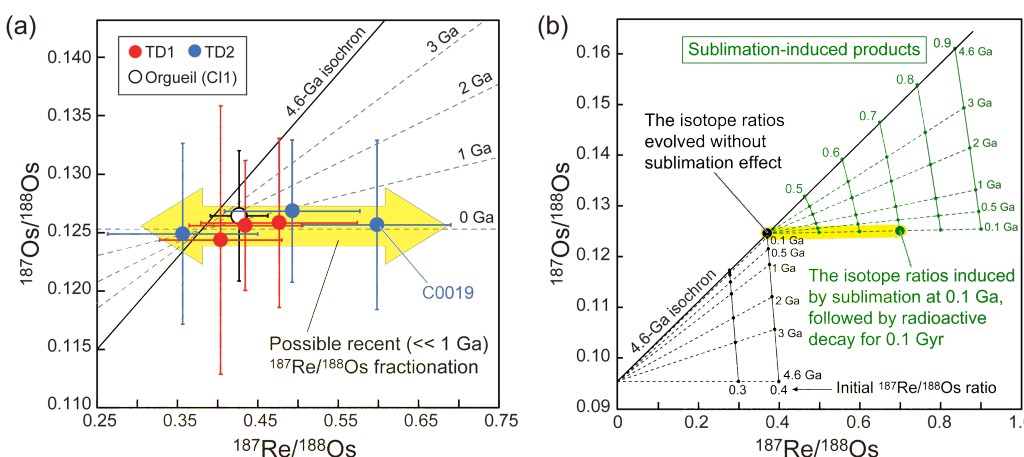

**Figure A1.** The Re and Os isotope diagrams. (**a**) The Re and Os isotope compositions of the Ryugu particles. The reference isochrons of 0, 1, 2, and 3 Ga are shown by broken lines for comparison. The Ryugu samples were plotted either on or higher than the 4.565 Ga isochron for $^{187}Re/^{188}Os$ [84] on the Re-Os isotope diagram [19]. The sample C0019 with the highest $^{187}Re/^{188}Os$ is characterized by not only Re but also Ir and Pd enrichments, whereas the μm-sized Fe sulfide crystal separated from the sample, which was regarded as a major reservoir of the highly siderophile elements (HSEs), was depleted in Re, Ir, and Pd [19]. Accordingly, nm-sized HSE-bearing compounds such as sulfides, oxides, and OM in the phyllosilicate-dominated matrix would have a high Re/Os ratio, which was acquired recently in geological terms. (**b**) A two-step isotope evolution model for the Re-Os isotope systematics of the Ryugu particles. Black broken lines indicate the first step of evolution since the formation of the solar system with the initial $^{187}Re/^{188}Os$ ratios of 0.3 and 0.4. The green broken lines indicate the second step of evolution after the sublimation. Numbers 0.5, 0.6, . . . 0.9, are the initial $^{187}Re/^{188}Os$ ratios of the HSE-bearing compounds formed by sublimation at 0.1, 0.5, . . . 4.6 Ga. In the case where the initial $^{187}Os/^{188}Os$ ratios of the Ryugu particles were 0.4 for both rocky and icy materials, and the sublimation-induced products with a $^{187}Re/^{188}Os$ of 0.7 were formed at 0.1 Ga, the present isotope ratios of the sublimation-induced products are shown as a solid green circle. The isotope ratios of the bulk Ryugu particles represent those of mixtures of the components with or without the sublimation effect, as shown by a yellow bar.

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
