# Peer review of "The Formation of a Rubble Pile Asteroid: Insights from the Asteroid Ryugu"

_universe, doi:10.3390/universe9060293_

Round 1
Reviewer 1 Report
The manuscript presents a short review on rubble pile asteroid formation focused on Ryugu. The text is very well presented and covers a significant scope of references.
I believe it brings some contribution in this topic to the community.
Author Response
We sincerely appreciate the reviewer’s careful reading our manuscript and providing the comments to improve the paper. As there was no specific comment for revi- sion from Reviewer 1 and Reviewer 2, we have modified the manuscript with considering each of the comments from Reviewer 3.
Reviewer 2 Report
I would first say that I am working on the area of asteroid exploring dynamical problems. So the evolution of asteroids/comets is related background of my research. This is also indicate that I am not very good at this scientific topic. Even though by going through the manuscript, I enjoyed reading this review paper. A new hypothesis was proposed regarding the rubble-pile asteroid based on the obversations of Ryugu. I would recommend its publication as a nice conceptual study. No specific comments.
Author Response

(The authors gave the same response as above.)

Reviewer 3 Report
Please see the attached file

Round 2
Reviewer 3 Report
I had a chance to give a look at the revised version of this manuscript. I appreciate that I am reviewing a very good paper with valuable results and am happy to see that the authors made an effort to accommodate all my previous comments. In my opinion, the paper in the current version is improved enough to become acceptable for publishing.